# Bridging Language Barriers in Smart Contract Security: A Reinforcement Learning-Based Translation Approach

## Abstract

Smart contracts have transformed the blockchain ecosystem, while flaws in their logic still present security risks. A significant disparity exists in the security tooling for smart contracts; while Solidity is well-supported, languages like Vyper and Rust lack robust vulnerability detection capabilities due to limited tools and data. To address this issue, we propose **RLTransSC**, a novel method that utilizes Reinforcement Learning (RL) to guide Large Language Models (LLMs) fine-tuning to translate smart contracts from low-resource languages into a high-resource language (i.e., Solidity). Once translated, we analyze the translated contracts using a vulnerability detection model trained specifically on Solidity. This approach removes the need for labeled datasets or specialized security tools for low-resource languages. We demonstrate the effectiveness of our method across Vyper, Rust, and Solidity, achieving strong translation performance and significantly improving vulnerability detection for low-resourced smart contract languages.

## 1 Introduction

Smart contracts have transformed decentralized applications. Solidity has emerged as the dominant language for Ethereum Babel et al. (2023); Sendner et al. (2024). Smart contract security is critical, as vulnerabilities can cause severe financial losses Hu et al. (2025); Chen et al. (2025b); Wu et al. (2025); Chen et al. (2025a). Tools like Mythril Sharma & Sharma (2022) have been effective for analyzing Solidity contracts. However, these tools largely overlook other languages like Vyper and Rust, which are increasingly adopted but lack mature tooling and datasets, posing significant challenges for vulnerability detection.

**Obstacles.** There are mainly two approaches to vulnerability detection in low-resourced smart contracts, both facing limitations. The *first* involves building *traditional program analysis tools* (e.g., symbolic execution), which requires extensive engineering to support each language. The *second* leverages *deep learning-based approaches*, but suffers from *data scarcity*—most non-Solidity languages lack sufficient labeled vulnerability datasets, making effective model training difficult. These challenges underscore the need for *cross-language vulnerability detection methods* that can effectively transfer knowledge between programming languages.

**Main Idea.** To address the data scarcity and engineering burden in language-specific vulnerability detection, we propose **RLTransSC**, a novel framework that *leverages reinforcement learning-guided LLM fine-tuning* to translate smart contracts from low-resource languages into a high-resource language (i.e., Solidity), enabling the application of mature security tools and rich labeled datasets. We then apply an existing Solidity vulnerability detection model to the translated code.

This method shows several advantages. *First*, it avoids building separate tools for each language. *Second*, it enables reuse of existing, high-performance detection models applied to code originating from other languages. *Third*, unlike traditional code translation methods that aim to improve code quality, our method explicitly preserves potential vulnerabilities, ensuring subsequent security analysis. *Finally*, by working within the Solidity ecosystem, we benefit from access to extensive tooling, pre-trained models, and curated vulnerability datasets that would otherwise be unavailable

in low-resource environments. The ***critical challenge*** lies in ensuring that *the translation preserves the original contract's semantics and security-relevant behavior* for vulnerability analysis.

**Approach.** Our approach leverages proximal policy optimization (PPO) to iteratively refine the LLM to translate smart contracts from low-resource languages into a high-resource language (i.e., Solidity). The RL formulation not only corrects subtle translation errors but also guides the LLM toward outputs that retain critical vulnerabilities in the translated code. Once trained, the model can translate contracts from low-resource languages (e.g., Vyper and Rust) into Solidity. The resulting Solidity contracts are then analyzed using existing vulnerability detection tools. This pipeline effectively bridges the gap between emerging smart contract languages and well-supported security tools by leveraging LLMs as a translation intermediary.

**Results.** We implement our approach, `RLTransSC`, to translate smart contracts from Vyper and Rust into Solidity, enabling the use of vulnerability detection tools in Solidity. We evaluate `RLTransSC` on four common vulnerability types: unauthorized transfer (UT), reentrancy (RE), arithmetic overflow (AO) and bad randomness (BR), using four state-of-the-art LLMs: CodeLlama-7b Roziere et al. (2023), CodeLlama-13b, StarCoder Li et al. (2023), and GPT-4.1 OpenAI (2025). Without RL-guided fine-tuning, baseline F1 scores are modest (53.4%-73.4%). After fine-tuning with parallel datasets, all model show substantial improvements. F1 scores increase to 92.0% for Vyper-translated contracts, and 93.5% for Rust-translated contracts. The results demonstrate the effectiveness of our RL-guided fine-tuning strategy in enhancing translation quality and enabling accurate vulnerability detection in low-resource languages. Below we highlight our contributions:

- We introduce `RLTransSC`, a novel framework that translates smart contracts from low-resource languages (e.g., Vyper and Rust) into Solidity, enabling the use of mature Solidity-based analysis tools without requiring language-specific analyzers.

- To support LLM-based translation, we construct a high-quality parallel dataset of semantically equivalent smart contracts and design a multi-faceted reward model for fine-tuning.

- We fine-tune four state-of-the-art LLMs using Proximal Policy Optimization (PPO) and show that our fine-tuned models outperform off-the-shelf versions in both translation accuracy and downstream vulnerability detection.

- Our end-to-end framework combines LLM-guided translation with existing vulnerability detection tools, achieving strong results across smart contract languages. We plan to make the source code, trained model and datasets publicly available.

## 2 RELATED WORK

**Vulnerability Detection in Smart Contracts.** In blockchain, despite the maturity of vulnerability detection tools for Solidity, support for low-resource languages like Vyper and Rust in blockchain remains limited. Static analysis tools such as Mythril Sharma & Sharma (2022) and Smartcheck Tikhomirov et al. (2018). Dynamic tools include Echidna Grieco et al. (2020), ContractFuzzer Jiang et al. (2018), and Harvey Wüstholz & Christakis (2020). Recently, deep learning models have gained traction in smart contract security Boi et al. (2024); Liu et al. (2024); Sendner et al. (2023); So et al. (2021); Liu et al. (2021); Sun et al. (2024). For Solana's Rust-based contracts, static tools include Radar Radar (2024) and VRust Cui et al. (2022b). Dynamic methods include FuzzDelSol Smolka et al. (2023). Deep learning models often require large labeled datasets He et al. (2024), which are scarce in low-resource languages. To mitigate this challenge, we propose *a novel translation-based framework that converts smart contracts from low-resource languages into Solidity, enabling the use of Solidity's mature security analysis tools.*

**LLM-based Code Translation.** Deep learning and Natural language processing have emerged as compelling tools Ma et al. (2024); Jiang et al.; Hou et al. (2025); Li et al. (2025); Chen et al. (2025b); Xu et al. (2025b); Wu & Arafin (2025); Xie et al. (2024); Zhang et al. (2024); Xu et al. (2025a); Sun et al. (2025); Zeng et al. (2025); Lin et al. (2025); Wang et al. (2025); Lan et al. (2025a). Popular Code LLMs include TransCoder Lachaux et al. (2020), StarCoder2 Li et al. (2023), CodeT5 Wang et al. (2021), and CodeLlama Roziere et al. (2023). Some are employing Reinforcement Learning (RL) to post-train LLMs like PPOCoder Shojaee et al. (2023), MAPPO Lan et al. (2025b), Co-Trans Jana et al. (2024), CodeRL Le et al. (2022), DPO Rafailov et al. (2023) and StepCoder Dou et al. (2024). Smart contract translation shows unique challenges compared with general code translation. The security-critical nature of smart contract demands that translations preserve not only

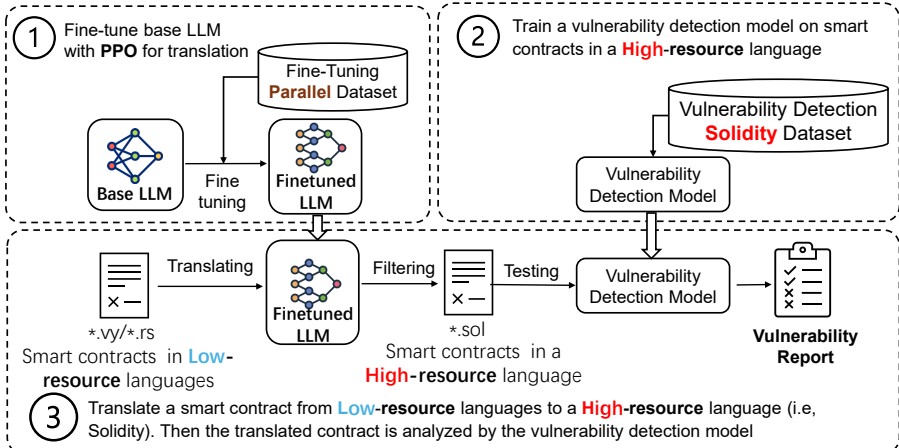

Figure 1: Applying **RLTransSC** to detect vulnerabilities in low-resource languages using a detection model targeting a high-resource language (e.g., Solidity).

functional behavior but also security properties. By framing smart contract translation as an RL problem, our approach-**RLTransSC** explicitly optimizes for vulnerability preservation while maintaining functional equivalence. This enables mature security tools to be applied to previously underserved smart contract languages.

## 3 MODEL DESIGN

### 3.1 OVERVIEW

Figure 1 presents an overview of **RLTransSC**. In step ①, we use PPO to fine-tune a base LLM to translate smart contracts from a low-resource language (i.e., Vyper and Rust) to a high-resource language (i.e., Solidity). In step ②, we train a deep learning-based vulnerability detection model using a *task-specific* Solidity dataset containing vulnerable and safe smart contracts. In step ③, when dealing with a contract in a low-resource language, we use the fine-tuned LLM to translate it to a high-resource language (i.e., Solidity) to detect vulnerabilities.

### 3.2 PARALLEL DATASET CONSTRUCTION

Translating smart contracts across programming languages poses unique challenges (i.e., requiring domain-specific knowledge), particularly when working with low-resource languages that are underrepresented in the training data of LLMs. To resolve the challenges, we construct a fine-tuning dataset that includes a curated collection of parallel contract pairs for both Vyper-Solidity and Rust-Solidity translation tasks. There are three steps for building a parallel fine-tuning dataset: (1) **Data collection**, (2) **Back translation** and (3) **Verification**.

**Data Collection.** Our data collection method involved harvesting smart contracts from multiple authoritative sources, including Etherscan, DeFiLlama, OpenZeppelin, DeFi protocols, and GitHub repositories. This ensures broad coverage of contract patterns, complexity levels, and domains.

**Back Translation.** To prevent the parallel dataset from overfitting to a single pattern, we employ multiple LLMs to generate synthetic parallel smart contracts. The details for using back-translation to generate synthetic parallel datasets are given in Appendix B.

**Verification.** To ensure semantic fidelity between generated candidates and source contracts, we developed an equivalence verification framework using *differential fuzzing*. Specifically, we implemented custom fuzzing harnesses using the Echidna Grieco et al. (2020) to conduct behavioral analysis of contract pairs. Our verification protocol involved feeding identical randomized inputs to both contract versions and performing comprehensive output validation, including: (1) return value equivalence, (2) event emission consistency and (3) state transition parity. For Rust-Solidity, we adapted our methodology to address the unique challenges posed by cross-platform blockchain translation. We developed additional platform-agnostic test harnesses that both Rust and Solidity implementations responded with semantically equivalent outcomes. Contracts successfully passing

our equivalence testing were incorporated into the parallel corpus, while those failing any verification criterion were excluded from the dataset. Our coverage-guided fuzzing campaigns achieve an average of 98.3% line coverage for the Vyper-Solidity corpus, and 98.9% coverage for Rust-Solidity.

We finally collected 1102 pairs for Vyper-Solidity and 1067 pairs for Rust-Solidity after deduplication and quality filtering. To provide the strongest possible evidence of semantic equivalence, we conduct real-world deployment testing on a random subset of our parallel dataset. We randomly select 50 pairs for Vyper-Solidity and 50 pairs for Rust-Solidity to deploy. For each deployed pair, we execute identical transaction sequences. And we observe an 97% consistency over state change, event emission and cross-platform behavior, which allows us to confidently assert the semantic equivalence of the parallel dataset.

### 3.3 MULTI-FACETED REWARD FUNCTION DESIGN

We model the smart contract translation task as a Markov Decision Process (MDP) $\mathcal{M} = \langle \mathcal{S}, \mathcal{A}, \mathcal{P}, \mathcal{R}, \gamma \rangle$ (See details in Appendix C). We design our reward function to address the fundamental challenge of preserving both *structural integrity* and *semantic equivalence* in smart contract translation through a carefully balanced combination of complementary objectives.

**AST Similarity Reward.** To enforce structural fidelity, we develop a reward that compares the Abstract Syntax Tree (AST) representations of the translated and ground-truth Solidity contracts. AST provides a canonical representation of grammatical structure, abstracting away superficial syntactical variations. We employ the Weisfeiler-Lehman (WL) graph kernel Shervashidze et al. (2011) to quantify AST similarity. There are ***two benefits of applying WL graph kernel*** to quantify similarity: (1) ***Flexibility in translation strategies.*** Exact AST isomorphism would overly constrain the translation process, forcing unnatural or suboptimal code structures. By allowing *partial similarity through WL kernel's feature composition*, the model can explore diverse translation approaches while maintaining essential structural integrity. (2) ***Preservation of Security-Critical Patterns.*** The WL kernel's iterative refinement naturally emphasizes structural patterns that are *crucial for vulnerability manifestation* (i.e., function call sequences for *unauthorized transfer*).

We adopt the Solidity compiler `solc` to extract ASTs from translated contract $G$ and groundtruth contract $\widehat{G}$. The normalized WL graph kernel similarity between $G$ and $\widehat{G}$ is computed as:

$$r_{\text{ast}} = \frac{\langle \phi_G, \phi_{\widehat{G}} \rangle}{\|\phi_G\|_2 \|\phi_{\widehat{G}}\|_2}$$

where $\phi_G$ and $\phi_{\widehat{G}}$ represent the feature vectors of generated AST and groundtruth AST. The normalized WL similarity score rewards translations that maintain security-relevant structural patterns without penalizing semantically equivalent but syntactically different implementations. So the LLM can explore diverse translation strategies while preserving the critical program properties.

**Test Case Execution Reward.** To ensure semantic equivalence, we employ a reward based on functional correctness through test case execution. We give five test cases against each translated Solidity contract and calculate the reward. This provides a direct, non-differentiable feedback of the translated code's behavioral fidelity. The reward is calculated as:

$$r_{\text{test}} = \begin{cases} -1.0 & \text{, if compilation fails} \\ -0.6 & \text{, if passing none unit tests} \\ +0.3 & \text{, if passing partial unit test} \\ +1.0 & \text{, if passing all unit tests} \end{cases}$$

**Composite Reward Function.** Our final reward function ($r : \mathcal{S} \times \mathcal{A} \rightarrow [-1, 1]$) that integrates two objectives (*structural fidelity* and *semantic faithfulness*) is designed as:

$$r(s, a) = \tanh(\alpha \cdot r_{\text{ast}} + \beta \cdot r_{\text{test}} - \lambda \cdot D_{\text{KL}}(\pi_\theta | \pi_{\text{ref}}))$$

where $\alpha$ and $\beta$ balance structural and functional objectives; $\lambda$ controls the regularization to prevent catastrophic forgetting; $D_{\text{KL}}(\pi_\theta | \pi_{\text{ref}})$ represents the KL divergence between the current policy $\pi_\theta$ and the pre-trained policy $\pi_{\text{ref}}$. This regularization term acts as a stability constraint, preventing the fine-tuning process from deviating too drastically from the base model's learned representations. The $\tanh$ function bounds rewards to $[-1, 1]$ for training stability. This multi-faceted reward design enables the model to learn translations that are both *structurally faithful* and *semantically correct* while maintaining the valuable knowledge encoded in the pre-trained foundation model.

## 3.4 PPO-BASED FINE-TUNING

We employ Proximal Policy Optimization (PPO) to fine-tune the base LLM using the reward function in Equation (3.3). PPO objective balances policy improvement with stability constraints:

$$\mathcal{L}^{\text{PPO}}(\theta) = \mathbb{E}_t \left[ \min \left( r_t(\theta)\widehat{A}, \text{clip}(r_t(\theta), 1 - \epsilon, 1 + \epsilon)\widehat{A} \right) \right] \tag{1}$$

where $r_t(\theta)$ is the probability ratio, $\widehat{A}$ is the advantage estimate computed as $\widehat{A} = r - V_\phi(s)$, and $\epsilon$ is the clipping parameter that constrains policy updates. The clipped surrogate objective prevents excessively large policy updates that could lead to performance degradation, ensuring that valuable knowledge from the pre-trained LLM is preserved.

**Training Procedure with LoRA Integration.** We integrate Low-Rank Adaptation (LoRA) with PPO to enable parameter-efficient fine-tuning. LoRA decomposes weight updates into low-rank matrices, significantly reducing trainable parameters while maintaining model expressiveness. The prompt for fine-tuning LLMs is given in Appendix D.

**Robust Candidate Generation.** To reduce LLM hallucination, we enhance our framework with an automated Generate, Verify, and Rank (GVR) module during inference. For each source contract, we generate 5 candidates, and each is graded on a composite score summed from two key metrics: *compilation integrity* $S_{\text{comp}}$ and *structural correspondence* $S_{\text{ast}}$. For *compilation integrity*, we calculate $S_{\text{comp}}$ by feeding the translated Solidity contract to compiler `solc`. Compilation success yields a score of 1, while a failure results in 0. This metric *penalizes severe LLM hallucinations* that produce non-executable code. For $S_{\text{ast}}$, we parse both the source contract and the candidate translation into ASTs using `tree-sitter` tree-sitter. Then we leverage GraphCodeBERT Guo et al. (2020) to generate feature vectors. $S_{\text{ast}}$ is measured by the cosine similarity between the source and candidate feature vectors. $S_{\text{ast}}$ rewards translations that faithfully replicate the logical blueprint of the original contract. The composite score is summed over $S_{\text{ast}}$ and $S_{\text{comp}}$. Then we select the candidate with the highest score. This approach provides a robust defense against LLM hallucination.

## 3.5 VULNERABILITY DETECTION

After fine-tuning the LLMs, we analyze translated contracts (from Vyper and Rust to Solidity) using existing vulnerability detection tools built for Solidity. This translation-based approach mitigates the data scarcity problem in low-resource languages. Our guidance for selecting vulnerability detection tools is that *a model must first demonstrate good performance on Solidity contracts*. If the model performs well on Solidity, it is more likely to also perform well on contracts translated from low-resource languages, assuming the translation is accurate. Following this guidance, we select two models: a deep learning–based model `AmeVulDetector` Liu et al. (2021) and an LLM-based model `GPTScan` Sun et al. (2024). In the following sections, we present the results of `AmeVulDetector`. Appendix H provides the implementations and results using `GPTScan`.

We adapted our vulnerability detection model based on `AmeVulDetector` Liu et al. (2021), which combines vulnerability-specific expert patterns with neural networks. The architecture consists of three main components: (1) Local expert pattern extraction, (2) Graph Construction, and (3) Attentive multi-encoder network. The implementation details are given in Appendix G.

## 4 EVALUATION

We conduct extensive experiments to assess **RLTransSC** in terms of translation accuracy and vulnerability detection effectiveness. Additionally, we perform hyperparameter studies.

**Research Questions.** To evaluate **RLTransSC**, we address three research questions (RQs):

---
**Research Questions**

**RQ1:** How effective is the translation performance of **RLTransSC**?
**RQ2:** What is the vulnerability detection performance of **RLTransSC** when applied to translate contracts across languages?
**RQ3:** How do different hyperparameter settings impact vulnerability detection performance?

---

Table 1: Translation evaluation results (%)

| Translation Model | Method | Vyper→Solidity | | | Rust→Solidity | | |
|---|---|---|---|---|---|---|---|
| | | CA@1 | CA@5 | CodeBLEU | CA@1 | CA@5 | CodeBLEU |
| CodeLlama 13b | Zero-shot | 23.1 | 34.6 | 20.6 | 17.1 | 32.6 | 26.4 |
| | Three-shot | 22.0 | 39.2 | 18.4 | 18.0 | 34.5 | 27.3 |
| | SFT-Full | 28.6 | 40.5 | 22.3 | 20.3 | 36.1 | 34.0 |
| | SFT-LoRA | 31.4 | 42.3 | 28.2 | 21.4 | 38.6 | 35.4 |
| | PPO-Full | 26.7 | 52.1 | 36.2 | 26.7 | 53.2 | 38.6 |
| | PPO-LoRA | **45.3** | **81.6** | **59.4** | **39.3** | **79.4** | **62.2** |
| CodeLlama 7b | Zero-shot | 26.0 | 32.2 | 32.4 | 20.3 | 31.3 | 35.1 |
| | Three-shot | 27.1 | 32.9 | 31.9 | 24.0 | 35.6 | 32.7 |
| | SFT-Full | 22.3 | 39.5 | 28.3 | 26.7 | 42.1 | 26.9 |
| | SFT-LoRA | 29.7 | 42.6 | 35.8 | 24.1 | 43.6 | 39.2 |
| | PPO-Full | 26.9 | 38.2 | 48.3 | 28.4 | 56.2 | 47.1 |
| | PPO-LoRA | **44.1** | **71.3** | **61.1** | **42.6** | **78.5** | **60.6** |
| StarCoder2 7b | Zero-shot | 17.6 | 38.1 | 38.6 | 12.9 | 29.4 | 34.1 |
| | Three-shot | 21.3 | 26.4 | 37.0 | 14.3 | 26.3 | 39.5 |
| | SFT-Full | 24.6 | 35.3 | 48.3 | 26.3 | 45.2 | 42.8 |
| | SFT-LoRA | 22.8 | 38.6 | 44.6 | 25.9 | 46.6 | 45.1 |
| | PPO-Full | 26.5 | 52.0 | 48.0 | 25.1 | 60.2 | 45.7 |
| | PPO-LoRA | **40.4** | **80.1** | **60.7** | **40.8** | **82.1** | **58.3** |
| GPT-4.1 | Zero-shot | 17.6 | 43.2 | 38.6 | 12.9 | 45.1 | 34.1 |
| | Three-shot | 21.3 | 46.5 | 37.0 | 14.3 | 50.9 | 39.5 |
| | SFT-Full | 24.6 | 57.2 | 48.3 | 26.3 | 59.4 | 42.8 |

## 4.1 RQ1: TRANSLATION EVALUATION

**Experimental Settings.** We employ three metrics to evaluate `RLTransSC`: (1) CA@1 (Computation Accuracy@1): The percentage of translated contracts that pass our differential testing on the first attempt; (2) CA@5: The percentage of contracts where at least one of the five translations passes our differential testing; and (3) CodeBLEU Ren et al. (2020): A popular metric that evaluates syntactic similarity, semantic equivalence, and structural alignment between the translated and ground-truth contracts. We selected four foundation models with varying sizes for RL-guided fine-tuning: (1) CodeLlama-7b, (2) CodeLlama-13b, (3) StarCoder2-7b, and (4) GPT-4.1. For the first three LLMs, we compare six different settings to provide a full analysis: (1) *Zero-shot learning*: The base LLM is prompted without any training examples or fine-tuning; (2) *Three-shot learning*: The LLM is provided with 3 random selected translation examples from the parallel dataset as in-context learning (ICL) prompts; (3) *SFT-Full*: Supervised fine-tuning with full parameter updating, where all model parameters are updated using standard maximum likelihood estimation; (4) *SFT-LoRA*: Supervised fine-tuning using Low-Rank Adaptation, updating a small subset of parameters; (5) *PPO-Full*: Proximal Policy Optimization with full parameter updating using our custom reward function; and (6) *PPO-LoRA*: PPO fine-tuning combined with LoRA for parameter-efficient training. Due to the limited fine-tuning settings provided by OpenAI, we thus include zero-shot learning, three-shot learning, and SFT-Full for GPT-4.1 into our evaluation.

**Results.** Our evaluation encompasses two pairs: Vyper-Solidity and Rust-Solidity. For Vyper-Solidity, our dataset comprises 1102 parallel contracts. For Rust-Solidity, we collect 1067 parallel contracts. We use 80% for training and 20% for testing for each pair. Table 1 shows our experimental results for four LLMs of varying architectural complexity and parameter scales for different settings. We include examples in Appendix F to show the effectiveness of vulnerability preservation.

**Findings.** The results reveal several important insights: (1) *Necessity of fine-tuning*: Fine-tuning surpasses other settings across all four LLMs, suggesting the importance of fine-tuning in improving translation performance. (2) *RL-guided fine-tuning superiority*: Fine-tuning LLMs with PPO demonstrates significant performance improvements compared to zero-shot learning, three-shot learning, and supervised fine-tuning across all metrics. (3) *LoRA effectiveness*: For the first three LLMs, the combination of PPO with LoRA (PPO-LoRA) consistently outperforms full parameter fine-tuning, suggesting that targeted parameter updates are more effective for this special-

ized translation task. (4) *Model capacity impact*: CodeLlama-13b significantly outperforms smaller alternatives, indicating that model capacity plays a crucial role in capturing the complex semantics.

> **Answer for RQ1:** RL-guided fine-tuning significantly enhances translation quality of LLMs, as evidenced by substantial improvements in CA@1, CA@5 and CodeBLEU across all LLMs.

### 4.2 RQ2: VULNERABILITY DETECTION RESULTS

In this section, we conduct experiments to answer **RQ2**. Our evaluation focuses on four critical vulnerability types: unauthorized transfer (UT), reentrancy (RE), arithmetic overflow (AO) and bad randomness (BR). UT occurs when a contract permits token transfers without authorization checks. RE arises when a function makes an external call, but the external contract maliciously calls back into the original function before the first invocation has completed, leading to unexpected state changes. It is noted that *while the mechanism differs for EVM and Solana* (external calls vs. cross-program invocations), *the core vulnerability pattern: state manipulation during external interactions applies across platforms*. Our translation preserves the logical structure that enables such attacks. AO occurs when arithmetic operations produce results that exceed their data type representation. BR denotes false use of on-chain data like `block.timestamp` to achieve randomness.

***Vulnerability Detection* Datasets and Evaluation Metric.** To build our *vulnerability detection* dataset, we collect vulnerable smart contracts from diverse sources, including Etherscan and GitHub repositories. For initial vulnerability labeling and classification, we employ established analysis tools including Slither Feist et al. (2019), Mythril Sharma & Sharma (2022), and Radar Radar (2024), which provide automated detection capabilities for known vulnerability patterns. Then, all labels undergo manual verification by the authors, who collectively possess extensive smart contract auditing experience. Finally, any contract with disputed labels is excluded from the dataset. For the evaluation metric, we assess vulnerability detection performance using F-1 score.

**Baseline: Vulnerability Detection *Without* Translation.** We first evaluate the performance of directly applying vulnerability detection tools to low-resource languages. This baseline comparison highlights the fundamental limitations of current methods and the need for our approach. We evaluate two categories of baselines: (1) *Deep learning-based model*: We adapt and extend `AmeVulDetector` to support Vyper and Rust via language-specific modifications. (2) *Traditional approaches*: We used three static tools: Slither Feist et al. (2019), Radar Radar (2024) and VRust Cui et al. (2022b) for comparison. Slither works on Vyper while Radar and VRust are for Rust. Additionally, we apply Mythril Sharma & Sharma (2022) as a symbolic execution tool on Vyper.

Table (2b) shows the results. For the *deep learning-based model*, we employ an 80/20 train-test split across each vulnerability category. The resulting F-1 scores are relatively low: 24.2%, 35.1%, 56.3% and 43.7% in Vyper, and 51.1%, 45.6%, 47.2% and 56.9% in Rust. This suboptimal performance stems primarily from the data scarcity issues inherent in low-resource languages: the lack of labeled vulnerable samples. While Solidity benefits from a wealth of resources (including vulnerability databases, security audit reports, and community-contributed datasets), Vyper and Rust suffer from extremely limited availability of labeled vulnerable samples. Despite our best efforts to collect and label smart contracts, the number of vulnerable samples we could gather for UT, RE, AO and BR vulnerabilities in Vyper and Rust remains significantly lower than in Solidity (see Appendix E).

For *traditional methods*, we obtained F-1 scores as high as 58.5%, 43.3%, 55.8% and 56.0% in Vyper, and 33.1%, 37.6%, 42.8% and 48.3% in Rust. These results reflect the limitations and relatively immature state of security toolchains for low-resource smart contract languages. Moreover, each language requires specialized preprocessing for unique features (e.g., Vyper's decorators, Rust's ownership semantics and Solana's account model), multiplying the engineering workload.

**Our Method: Vulnerability Detection *With* Translation.** Our innovative translation-based approach, which systematically translates smart contracts from low-resource languages into Solidity before applying vulnerability detection, offers several significant advantages: (1) It leverages Solidity's rich ecosystem and extensive labeled datasets. (2) It mitigates data scarcity by enabling unified vulnerability detection in a well-resourced target language. (3) It preserves vulnerability semantics while benefiting from mature analysis tools available in Solidity.

Table 2: Vulnerability Detection Results (%)

(a) Vulnerability Detection *With* Translation

| Translation Model | Method | Vyper | | | | Rust | | | |
|---|---|---|---|---|---|---|---|---|---|
| | | UT | RE | AO | BR | UT | RE | AO | BR |
| CodeLlama 13b | Zero-shot | 64.7 | 73.4 | 72.1 | 64.5 | 69.2 | 71.8 | 72.6 | 65.0 |
| | Three-shot | 59.5 | 68.3 | 67.2 | 66.8 | 64.1 | 64.9 | 70.4 | 62.6 |
| | SFT-Full | 72.9 | 76.2 | 75.3 | 73.7 | 74.5 | 77.4 | 77.0 | 69.1 |
| | SFT-LoRA | 72.8 | 74.5 | 76.1 | 76.0 | 75.4 | 77.7 | 78.2 | 71.9 |
| | PPO-Full | 80.1 | 80.9 | 84.6 | 86.7 | 87.3 | 85.8 | 85.1 | 90.2 |
| | PPO-LoRA | **88.5** | **89.4** | **89.7** | **92.0** | **94.3** | **88.9** | **89.1** | **93.8** |
| CodeLlama 7b | Zero-shot | 59.2 | 71.1 | 70.4 | 58.9 | 68.7 | 67.6 | 70.3 | 61.0 |
| | Three-shot | 71.6 | 71.7 | 71.5 | 68.1 | 62.4 | 69.9 | 73.8 | 67.3 |
| | SFT-Full | 72.4 | 73.6 | 76.3 | 76.7 | 69.8 | 72.1 | 76.9 | 70.8 |
| | SFT-LoRA | 78.5 | 80.4 | 82.7 | 80.8 | 75.6 | 75.3 | 76.4 | 71.7 |
| | PPO-Full | 85.3 | 83.9 | 84.4 | 85.1 | 88.5 | 85.2 | 85.9 | 90.6 |
| | PPO-LoRA | **86.4** | **87.8** | **87.3** | **91.5** | **92.9** | **87.2** | **87.7** | **91.4** |
| StarCoder2 7b | Zero-shot | 62.2 | 58.6 | 59.9 | 64.1 | 58.7 | 54.2 | 53.4 | 60.8 |
| | Three-shot | 60.1 | 59.3 | 55.8 | 63.5 | 57.4 | 55.0 | 52.6 | 59.2 |
| | SFT-Full | 60.5 | 62.7 | 51.8 | 65.9 | 68.3 | 59.1 | 57.2 | 61.7 |
| | SFT-LoRA | 66.9 | 70.2 | 68.4 | 68.5 | 65.7 | 61.9 | 63.5 | 71.6 |
| | PPO-Full | 67.4 | 75.6 | 70.7 | 72.3 | 67.8 | 65.4 | 73.5 | 86.9 |
| | PPO-LoRA | **86.6** | **86.5** | **85.2** | **89.8** | **89.4** | **87.1** | **86.7** | **93.5** |
| GPT-4.1 | Zero-shot | 82.4 | 81.9 | 82.6 | 83.3 | 84.5 | 82.1 | 77.8 | 82.7 |
| | Three-shot | 77.6 | 81.4 | 82.7 | 76.2 | 84.9 | 80.5 | 75.4 | 81.6 |
| | SFT-Full | 84.8 | 84.2 | 83.5 | 85.4 | 87.1 | 85.3 | 83.6 | 89.2 |

(b) Vulnerability Detection *Without* Translation

| Method | Language | Vyper | | | | Rust | | | |
|---|---|---|---|---|---|---|---|---|---|
| | | UT | RE | AO | BR | UT | RE | AO | BR |
| AmeVulDetector | Vyper | 24.2 | 35.1 | 56.3 | 43.7 | - | - | - | - |
| | Rust | - | - | - | - | 51.1 | 45.6 | 47.2 | 56.9 |
| Slither | Vyper | 58.5 | 43.3 | 55.1 | 56.0 | - | - | - | - |
| Mythril | Vyper | 47.4 | 39.8 | 55.8 | 50.6 | - | - | - | - |
| Radar | Rust | - | - | - | - | 33.1 | 37.6 | 42.8 | 48.3 |
| VRust | Rust | - | - | - | - | 28.1 | 43.9 | 32.5 | 35.7 |

We first fine-tune LLMs using PPO on our parallel dataset, then translate a smart contract in a low-resource language to a Solidity version. Then we apply `AmeVulDetector` trained on Solidity to analyze the translated contracts. Specifically, we train a separate model for each of the four vulnerability types. Using unauthorized transfer (UT) as an example: our dataset comprises 883 safe and 452 vulnerable samples. We employ an $80/20$ train-test split, resulting in a training set of 706 safe and 362 vulnerable samples, and a test set of 177 safe and 90 vulnerable samples. We follow the same training methodology for other three vulnerability types.

**Results.** Table 2 presents the detailed vulnerability detection results for four vulnerability types using translation LLMs with different settings. Specifically, our approach achieves F-1 scores up to 88.6.%, 89.4%, 89.7% and 92.0% for UT, RE, AO and BR for Vyper. For Rust, we achieve F-1 scores of 94.3%, 88.9%, 89.1% and 93.8%. These consistently high-performance metrics across different vulnerability types, programming languages, and model architectures clearly demonstrate the effectiveness and robustness of our **RLTransSC** framework. Additionally, we discuss the transferability of smart contract vulnerability in Appendix I.3.

To establish a performance baseline, we first evaluate `AmeVulDetector` when trained and tested exclusively on Solidity smart contracts *without any translation*. We maintain the same $80/20$ train-test split across each vulnerability type on our Solidity dataset. We achieve F-1 scores of 95.1%, 95.4%, 93.2% and 95.5% for UT, RE, AO and BR. Next, we evaluate the same Solidity-trained models on smart contracts translated from Vyper and Rust to demonstrate the applicability of our translation-based approach. Most importantly, the vulnerability detection models remain completely unchanged throughout evaluation. When comparing our approach to direct Solidity analysis, we

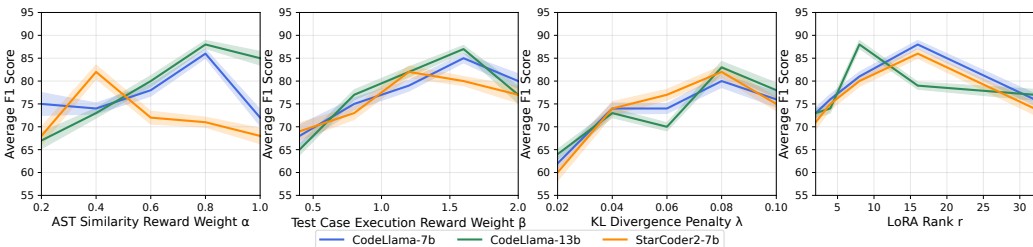

Figure 2: Hyperparameter study results averaged over four vulnerability types.

observe only minimal performance decreases. Specifically, our approach achieves F-1 scores of $86.4\%$, $87.8\%$, $87.3\%$ and $91.5\%$ for UT, RE, AO and BR on contracts translated from Vyper using CodeLlama-7b, representing decreases of only $8.7\%$, $7.6\%$, $5.9\%$ and $4.0\%$ compared to the native Solidity performance. This demonstrates that **RLTransSC** successfully enables effective vulnerability detection for underserved languages while maintaining competitive detection capabilities with minimal performance degradation. Appendix H.1 presents examples illustrating how vulnerabilities are preserved through LLM-powered translation.

> **Answer for RQ2: RLTransSC** achieves F-1 scores up to $92.0\%$ for Vyper and $94.3\%$ for Rust contracts, demonstrating effective vulnerability detection performance.

### 4.3 RQ3: HYPERPARAMETER STUDY

We study the impacts of four hyperparameters-AST similarity reward weight $\alpha$, Test case execution reward weight $\beta$, KL divergence penalty $\lambda$ and LoRA rank $r$ on the vulnerability detection performance. We report the average F-1 score for each of the four vulnerability types, fine-tuned for each LLM. The complete results are shown in Figure 2.

**AST similarity reward weight.** For all models, performance is poor at a low $\alpha$ (0.2), the LLM fails to generate syntactically valid translations. Performance slightly declines when $\alpha$ is too high (1.0) because an excessive focus on matching the ground-truth AST can make the model overly rigid.

**Test case reward weight.** $\beta$ provides a direct signal of functional correctness. Performance for all models increases monotonically with $\beta$. The performance begins to plateau after $\beta = 1.2$, with the optimal value for all models being $1.5$. A strong $\beta$ value is essential for guiding the model to produce semantically faithful translations.

**KL divergence penalty.** $\lambda$ regularizes the fine-tuning process, preventing the policy from deviating too far from the base model. When $\lambda$ is low (0.02), performance suffers due to *catastrophic forgetting*. Conversely, when $\lambda$ is too high (0.1), it prevents the model from learning the new task effectively. The optimal value for all LLMs is found at $\lambda = 0.08$.

**LoRA Rank.** $r$ determines the number of trainable parameters. We evaluated ranks from 4 to 32. We can see for CodeLlama-13b, the performance peaks when $r = 8$. For CodeLlama-7b and StarCoder2-7b, the performance peaks at $r = 16$. This is expected since, given the same amount of dataset, a larger model can achieve optimal performance within a smaller set of updating parameters.

> **Answer for RQ3: RLTransSC**'s vulnerability detection performance is sensitive to hyperparameters. The optimal values vary with each specific LLM.

## 5 CONCLUSION

In this work, we introduce **RLTransSC**, a novel framework that addresses the security analysis limitations of low-resource smart contract languages by leveraging LLM-based translation. **RLTransSC** converts smart contracts written in Vyper and Rust into Solidity, enabling the use of mature and widely adopted analysis tools within the Solidity ecosystem to assess contracts originally developed in less-supported languages. To ensure high-quality translations, we fine-tune LLMs on carefully curated parallel datasets using PPO. This enables **RLTransSC** to retain both functionality and potential security flaws in the translated code. Our vulnerability detection pipeline that integrates AmeVulDetector and GPTScan exhibits strong performance.

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

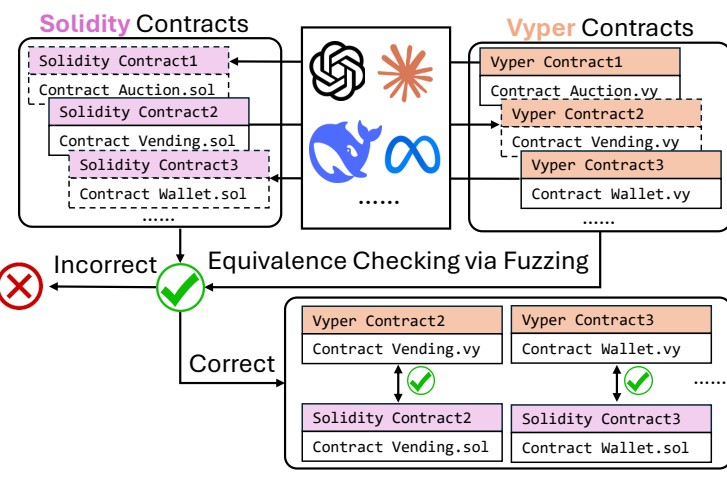

Figure 3: Use back-translation to generate synthetic parallel datasets.

## A  COMPUTING RESOURCES

All experiments were performed in a high-performance computing environment to ensure efficient training and evaluation of large language models. Specifically, we utilized a dedicated server running Ubuntu 20.04, equipped with an Intel Xeon Platinum 8358 CPU (32 cores, 64 threads) and two NVIDIA A800 GPUs, each with 80 GB of VRAM. This infrastructure provided the computational capacity required for large-scale model training and comprehensive evaluation across multiple model variants.

## B  PARALLEL DATASET BUILDING DETAILS

Figure 3 shows the process of building parallel datasets using multiple LLMs. We use multiple types of LLMs to generate synthetic translation candidates, including GPT-4.1, Llama-3.2, Deepseek-R1, Claude-Sonnet-4, Gemini-2.5-Flash. Take Vyper-Solidity as an example, for a Solidity contract in our collection, we employed each LLM to produce 3 candidate parallel implementations in Vyper using each prompt template. We applied the symmetric approach for Vyper contracts, generating Solidity equivalents. All generated candidates go through the verification step.

## C  MDP DESIGN

We model the smart contract translation task as a Markov Decision Process (MDP) $\mathcal{M} = \langle \mathcal{S}, \mathcal{A}, \mathcal{P}, \mathcal{R}, \gamma \rangle$ to enable reinforcement learning-guided fine-tuning. We explain the components of MDP as follows:

- State Space $\mathcal{S}$: Each state $s_t$ represents the current partial translation context at a given time step $t$. It consists of the complete source code and the sequence of target tokens generated thus far, $s_{t+1} = (\text{source\_code}, y_1, \cdots, y_t)$

- Action Space $\mathcal{A}$: The action space is the discrete vocabulary of the target language (Solidity). An action $a_t$ corresponds to the LLM selecting the next token $y_t$ to append to the translation sequence.

- Transition Function $\mathcal{P}$: Deterministic transitions based on the language model's token generation mechanism. Given a state $s_t$ and an action $a_t$, the next state is deterministically $s_{t+1} = (\text{source\_code}, y_1, \cdots, y_t)$

- Reward Function $\mathcal{R}$: The reward is a scalar feedback signal that evaluates the quality of a completed translation. In our setting, the reward is sparse and delayed. it is only provided at the end of an episode (i.e., after the entire contract has been generated).

- Policy $\pi_\theta$: The policy is the LLM parameterized by weights $\theta$, mapping states to probability distributions over the action space-$\pi_\theta(a_t|s_t)$, defining the likelihood of generating possible next token.

- Discount Factor $\gamma$: Set to 1.0 since rewards are only given for the final, complete translation.

## D FINE-TUNING PROMPT

For fine-tuning base LLMs using PPO, we structured the data using our designed prompt template (Figure 4) to ensure consistent instruction formatting.

---

**Prompt to Translate Smart Contracts**

**System:** You are a smart contract translation expert fluent in both `[SRC_LANG]` and `[TGT_LANG]`. Your task is to translate code written in `[SRC_LANG]` into equivalent code in `[TGT_LANG]`, adhering to the following requirements:
1. The translated contract must maintain the same logic, including all events emitted, state changes, and blockchain-specific operations.
2. If the source contract contains security vulnerabilities, they must be retained in the translated contract.
3. All function and variable names must remain unchanged.
Input `[SRC_LANG]`: `[SRC_CONTRACT]`
Output: `[TGT_LANG]`: `[TGT_CONTRACT]`

---

Figure 4: Prompt for translation task.

## E DATASET

Table 3 presents the statistics of our *task-specific* dataset.

| Language | Safe | UT | RE | AO | BR |
|----------|------|-----|-----|-----|-----|
| Solidity | 883 | 452 | 383 | 492 | 423 |
| Vyper | 234 | 136 | 126 | 128 | 148 |
| Rust | 285 | 165 | 146 | 133 | 124 |

Table 3: Statistics for *task-specific* dataset.

## F TRANSLATION DEMONSTRATION

In Figure 5, we give two examples of translation pairs: Vyper-Solidity and Rust-Solidity to show the effectiveness of our fine-tuned LLM.

## G IMPLEMENTATION DETAILS OF AMEVULDETECTOR

This appendix details the design and implementation of `AmeVulDetector` for detecting the four vulnerability types covered in our evaluation: unauthorized transfer (UT), reentrancy (RE), arithmetic overflow (AO), and bad randomness (BR). Here we give the expert pattern design of four vulnerabilities in Table 5. For the graph construction, we build the node identification as follows.

**Node Identification.** We design three node identifications for each vulnerability type: (1) UT: Authorization check functions; Transfer function invocations; Balance-related variables and mappings.

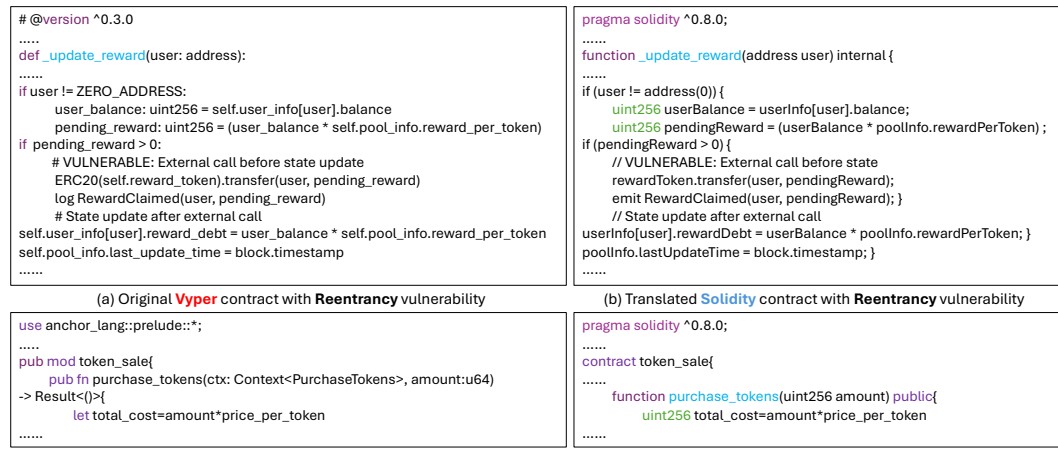

Figure 5: Examples of translation

(2) RE: External call sites; Balance variables and state modifications; Fallback function references. (3) AO: Arithmetic operation sites; SafeMath function calls; Boundary check statements. (4) BR: Block property accesses; Random number generation functions; Critical decision points using randomness.

**Training Configuration.** We give the key hyperparameters used for training `AmeVulDetector` in Table 4.

| Parameter | Value | Description |
|---|---|---|
| **Graph Neural Network** | | |
| Loss Func. | - | Weighted binary cross-entropy |
| GNN Layers | 3 | Number of message passing layers |
| Hidden Dimension | 128 | Node embedding dimensions |
| Aggregation Func. | Mean+Max | Graph-level feature aggregation |
| Message Passing | GRU | Update function for node states |
| **Attention Mechanism** | | |
| Attention Heads | 16 | Number of attention heads per layer |
| Attention Dropout | 0.1 | Dropout in attention layers |
| Self-Attention Dim. | 64 | Self-attention hidden dimensions |
| Cross-Attention Dim. | 128 | Number of attention heads per layer |
| **Expert Pattern Encoding** | | |
| Pattern Embedding. | 32 | Dimension for pattern features |
| Pattern MLP Layers | 2 | Layers in pattern processing MLPs |
| Pattern MLP Hidden | 64 | Hidden units in pattern MLPs |
| **Feature Fusion** | | |
| Fusion Type | Concatenation | Method to combine features |
| Fusion Hidden Dim. | 256 | Hidden dimensions in fusion layer |
| Fusion Layers | 3 | Number of fusion MLP layers |

Table 4: Parameter Details of `AmeVulDetector`.

# H IMPLEMENTATION DETAILS OF GPTSCAN

As a complementary work of vulnerability detection model `AmeVulDetector`, we adapt `GPTScan` Sun et al. (2024), which combines LLM reasoning with static analysis to detect logical vulnerabilities in Solidity. `GPTScan` adopts an innovative approach of breaking down vulnerabilities into code-level scenarios and properties. This decomposition enables direct matching of candidate functions with specific vulnerability patterns at the semantic level.

| | Expert Pattern Design |
|---|---|
| Unauthorized Transfer (UT) | **AuthCheck**: Detect authorization verification before transfer operations
**TransferFunction**: Identify function that perform token transfers.
**PrivelegeEscalation**: Detect potential privilege escalation vectors. |
| Reentrancy (RE) | **EnoughBalance**: Checks for balance verification before transfers.
**BalanceDeduction**: Verifiy that balance updates occur before external calls.
**ExternallCall**: Detect external call operations that could trigger reentrancy. |
| Arithmetic Overflow (AO) | **ArithmeticOperation**: Identify arithmetic operations susceptible to overflow.
**SafeOperations**: Check for usage of overflow protection mechanisms.
**BoundaryCheck**: Detect boundary validation before arithmetic operations. |
| Bad Randomness (BR) | **RandomnessSource**: Identify usage of weak randomness sources.
**RandomnessUsage**: Check if weak randomness affects critical operations.
**ExternalRandomness**: Detect usage of external randomness oracles. |

Table 5: Implementation of `AmeVulDetector`.

| | Scenario and Property | Filtering Type | Static Check |
|---|---|---|---|
| UT | **Scenario**: Functions that involve transferring tokens from an address different from the message sender, typically through transferFrom, safeTransferFrom, or similar token transfer mechanisms.
**Property**: The function lacks proper authorization checks such as allowance verification, approval validation, or ownership confirmation before executing the transfer. | FNK, FCNE, FCE, FPNC | VC, FA, DF |
| RE | **Scenario**: Functions that make external calls to other contracts or addresses, particularly through low-level calls, token transfers with callbacks, or cross-contract interactions.
**Property**: The function modifies contract state after making external calls, creating opportunities for malicious contracts to recursively call back before state updates are complete. | FCE, FCCE, FNM | OC, DF, VC |
| AO | **Scenario**: Functions performing arithmetic operations such as addition, multiplication, subtraction, or division on user-controlled inputs or contract variables.
**Property**: Arithmetic operations lack proper bounds checking, overflow protection, or safe math libraries, potentially causing integer overflow/underflow. | FCE, FCNE, FCCE | VC, DF, FA |
| BR | **Scenario**: Functions that require randomness for security-critical operations such as lottery selection, random rewards distribution, or unpredictable value generation.
**Property**: The function uses predictable sources of randomness such as block.timestamp, block.difficulty, block.number, or blockhash instead of secure randomness sources. | FNK, FCE, FCCE | FC, DF, FA |

Table 6: Breaking down four vulnerability types into scenarios and properties.

**Limitations of Direct Applying GPTScan.** `GPTScan` is tightly coupled with Solidity, making direct adaptation to low-resource languages impractical due to several challenges: (1) Language-specific semantics – Vyper's Pythonic syntax, decorators (e.g., `@external`, `@payable`), and built-in checks require redefining vulnerability scenarios. (2) Toolchain dependency – `GPTScan` relies on Solidity-based static analysis tools, which lack equivalents for Vyper or Rust (e.g., symbolic execution, CFG/data flow analyzers). (3) Platform divergence – Rust on Solana introduces architectural challenges like PDAs and CPIs, which have no analogs in Ethereum-based languages, necessitating an entirely new detection logic.

**Advantage of GPTScan.** By translating low-resource smart contracts to Solidity, we sidestep the need to reengineer `GPTScan` for each smart contract language. This allows us to leverage `GPTScan`'s mature vulnerability scenarios and analysis capabilities immediately. Our approach combines the strengths of LLM-based translation with powerful detection, enabling accurate vulnerability analysis for previously underserved smart contract languages with minimal engineering effort.

**Vulnerability Breakdown.** We break down the four vulnerability types into scenarios and properties shown in Table 6. For the GPT model configuration, we use the `GPT-4-turbo` model with temperature 0. The context is $4,000$ tokens, which is sufficient for filtering. And we follow the `GPTScan`'s "mimic-in-the-background" prompting technique for consistency.

**Results.** We give the complete results of vulnerability detection using an LLM-based detection model-`GPTScan`, The results are shown in Table 7. The results show that our approach achieves F-1 scores up to 87.2, 89.6, 88.4 and 92.1 for UT, RE AO and BR for Vyper. For Rust, we achieve F-1 scores of 93.3, 88.7, 88.6 and 92.5 for UT, RE, AO and BR after translation. These high-performance metrics show the effectiveness of our **RLTransSC** framework and generalization to different vulnerability detection tools.

| Translation Model | Method | Vyper | | | | Rust | | | |
|---|---|---|---|---|---|---|---|---|---|
| | | UT | RE | AO | BR | UT | RE | AO | BR |
| CodeLlama-13b | Zero-shot | 63.5 | 72.9 | 71.6 | 64.7 | 68.8 | 70.8 | 71.4 | 64.5 |
| | Three-shot | 58.4 | 68.2 | 67.1 | 66.2 | 63.4 | 64.6 | 69.4 | 62.0 |
| | SFT-Full | 71.3 | 76.1 | 74.3 | 73.1 | 74.6 | 76.5 | 77.3 | 69.1 |
| | SFT-LoRA | 72.7 | 74.4 | 76.0 | 75.7 | 74.8 | 77.4 | 78.0 | 70.5 |
| | PPO-Full | 79.8 | 80.7 | 84.4 | 86.1 | 87.4 | 84.3 | 85.5 | 90.1 |
| | PPO-LoRA | **87.2** | **89.6** | **88.4** | **92.1** | **93.3** | **88.7** | **88.6** | **92.5** |
| CodeLlama-7b | Zero-shot | 69.1 | 73.8 | 72.3 | 68.1 | 68.4 | 71.1 | 71.7 | 65.5 |
| | Three-shot | 71.7 | 70.7 | 71.0 | 67.8 | 61.7 | 69.5 | 72.5 | 66.8 |
| | SFT-Full | 71.8 | 72.8 | 75.2 | 76.1 | 69.2 | 71.3 | 75.9 | 69.2 |
| | SFT-LoRA | 78.5 | 80.1 | 82.7 | 79.2 | 74.9 | 75.4 | 75.9 | 70.8 |
| | PPO-Full | 85.5 | 82.8 | 84.5 | 85.0 | 88.4 | 85.1 | 85.4 | 89.6 |
| | PPO-LoRA | **86.0** | **87.5** | **86.2** | **90.5** | **91.2** | **87.2** | **87.1** | **90.3** |
| StarCoder2-7b | Zero-shot | 62.1 | 58.4 | 59.0 | 63.7 | 57.8 | 54.1 | 52.6 | 59.8 |
| | Three-shot | 59.4 | 58.9 | 54.3 | 62.4 | 56.9 | 55.1 | 51.3 | 59.6 |
| | SFT-Full | 60.6 | 61.4 | 51.6 | 64.7 | 68.5 | 58.8 | 57.0 | 60.3 |
| | SFT-LoRA | 65.3 | 70.4 | 68.1 | 68.6 | 65.1 | 60.4 | 62.5 | 70.2 |
| | PPO-Full | 67.0 | 75.4 | 70.0 | 71.8 | 67.6 | 64.8 | 72.3 | 85.4 |
| | PPO-LoRA | **85.9** | **86.2** | **84.4** | **89.1** | **89.7** | **87.1** | **85.7** | **92.2** |
| GPT-4.1 | Zero-shot | 84.1 | 78.6 | 80.9 | 82.1 | 83.0 | 81.2 | 76.3 | 80.5 |
| | Three-shot | 77.8 | 82.2 | 83.1 | 77.0 | 86.7 | 78.8 | 75.7 | 81.9 |
| | SFT-Full | 83.1 | 81.6 | 82.9 | 82.1 | 84.5 | 83.2 | 78.3 | 81.7 |

Table 7: Vulnerability Detection Results using `GPTScan`

## H.1 CASE STUDY: RUSTFUND REENTRANCY VULNERABILITY

To validate the practical efficacy and real-world applicability of the **RLTransSC** framework, we present a comprehensive case study on a high-impact vulnerability from the Solana ecosystem. We apply the full **RLTransSC** pipeline to demonstrate its end-to-end capability in addressing security flaws in low-resource languages. The case study follows the three primary stages outlined in our methodology: (1) Vulnerability Identification : We analyze a known Common Vulnerability, deconstruct its fundamental weakness, and formulate a representative smart contract in a low-resource language (Rust) that exhibits the same class of vulnerability. (2) Vulnerability-Preserving Translation: We employ our fine-tuned LLM to translate the vulnerable Rust contract into its semantic equivalent in Solidity, with the explicit goal of preserving the security flaw. (3) Detection in High-Resource Environment: We use the two Solidity-based detection models proposed in our framework, `GPTScan` and `AmeVulDetector`, to analyze the translated code and confirm the detection of the preserved vulnerability.

**Contextualizing the Vulnerability.** The vulnerability selected for this case study is the RustFund reentrancy risk RustFund (2023), a cross-program invocation (CPI) reentrancy vulnerability that affected multiple Solana DeFi protocols in 2023. This vulnerability occurs when a Solana program makes external calls to other programs without proper state management, allowing malicious programs to re-enter the calling function before the original execution completes. Unlike traditional Ethereum reentrancy that exploits external contract calls, Solana reentrancy exploits the cross-program invocation mechanism, making it a platform-specific vulnerability pattern that demonstrates the importance of cross-language security analysis.

**Translation Process.** We utilize the fine-tuned CodeLlama-13b model for translation, which demonstrated superior performance in our evaluation. The translation process is guided by the specialized prompt template, as shown in Figure 4, which explicitly instructs the model to maintain logical consistency and, most importantly, to retain any security vulnerabilities present in the source contract. Figure 6(b) presents the Solidity contract translated from the original Rust contract

```
use anchor_lang::prelude::*;
use anchor_spl::token::{self, Token, TokenAccount, Transfer};
pub mod fund {
    use super::*;
    ......
    pub fn withdraw(ctx: Context<Withdraw>) -> Result<()> {
        let amount = ctx.accounts.vault.balance;
        // Transfer funds first
        **ctx.accounts.vault. try_borrow_mut_lamports()? -= amount;
        **ctx.accounts.creator. try_borrow_mut_lamports()? += amount;
        // Update state after transfer (vulnerable)
        ctx.accounts.vault.balance = 0;
}
......
```

(a) Original **Rust** contract with **RustFund**
**Reentrancy** vulnerability

```
// SPDX-License-Identifier: MIT
pragma solidity ^0.8.0;
Contract Fund{
    IERC20 public token;
    ......
    function withdraw() public {
        uint256 amount = vault.balance;
        payable(msg.sender).transfer(amount);
        vault.balance = 0;
    }
......
```

(b) Translated **Solidity** contract with
**Reentrancy** vulnerability

**GPTScan:**
**Vulnerability Type**: Reentrancy Attack
**Severity**: HIGH
**Confidence**: 97.1%
**Description**: The function performs an external call to token.transfer()
before updating the contract state (userBalance and totalBalance). This
creates a reentrancy vulnerability where a malicious token contract
could call back into withdraw() before the balance updates are
completed.
**Recommendation**:
1. Implement checks-effects-interactions pattern
2. Use ReentrancyGuard modifier
3. Update state before external calls

(c) Vulnerability detection result
using `GPTScan`

**AmeVulDetector:**
**Vulnerability Type**: Reentrancy
**Confidence Score**: 98.7%
**Location**: withdraw() function, line 28-35
**Pattern Matched**: External call followed by
state modification without reentrancy
protection

(d) Vulnerability detection result using
`AmeVulDetector`

Figure 6: A case study demonstrating the detection of a vulnerability in a Rust smart contract through cross-language translation.

in Figure 6(a), demonstrating that the vulnerability present in the original contract is preserved in the translated version.

**Results and Analysis.** After translation, we apply both `AmeVulDetector` and `GPTScan` to analyze the translated Solidity contract. The results are shown in Figure 6(c) and (d), respectively. Both detection models successfully identify the reentrancy vulnerability with a high confidence level (98.7% and 97.1%). `GPTScan` reported the vulnerable function `withdraw()` in the translated contract and reported the severity level as high. For `AmeVulDetector`, it matches the pattern of external call followed by state modification without reentrancy protection and locates the location of the vulnerability.

This case study validates the core design principles of **RLTransSC** : Vulnerability preservation across platforms. The translation successfully preserves the fundamental reentrancy pattern despite the significant architectural differences between Solana's CPI mechanism and Ethereum's external contract calls. The vulnerable sequence, including balance check, external call, state update, remains intact in both versions, demonstrating **RLTransSC**'s ability to maintain security-critical logical structures.

# I   DISCUSSION

## I.1   UNIQUENESS OF OUR WORK

This work represents the first approach to vulnerability detection in low-resource smart contract languages through a translation-based methodology. While prior work has explored cross-language smart contract translation, our approach addresses a fundamentally different challenge with distinct requirements. We aim to explicitly preserve vulnerabilities during translation, which is essential for subsequent vulnerability detection.

Our vulnerability-preserving, documentation-independent approach to multi-language smart contract translation represents a novel contribution to blockchain security analysis. By maintaining the integrity of security-relevant patterns while enabling cross-language analysis, our work fills a critical gap in the detection of vulnerabilities across diverse smart contract platforms.

## I.2  WHY NOT BYTECODE?

Since both Solidity and Vyper ultimately compile to EVM bytecode for execution on the Ethereum Virtual Machine, a seemingly natural and straightforward question arises: why not perform vulnerability analysis directly at the bytecode level rather than at the source code level? In this way, a tool designed to analyze Solidity bytecode should also be applicable to Vyper bytecode, given their shared compilation target. However, this approach faces several significant challenges.

**Toolchain Variability.** The compilation process itself introduces inherent complexity. Solidity uses `solc` for compilation while Vyper uses `Vyperlang`. Furthermore, different compiler versions, language-specific abstractions and optimization settings can produce widely varying bytecode for identical source code. This variability further complicates attempts to develop universal bytecode-level detection patterns that work reliably across languages and compiler configurations Chen et al. (2020); Albert et al. (2018).

**Cross-Platform Incompatibility.** The challenge is even more pronounced for Rust contracts on Solana, which operate on an entirely different blockchain architecture with fundamentally different execution models, account structures, and state management paradigms Smolka et al. (2023). Rust is capable of being compiled to eBPF bytecode. The semantic gap between EVM bytecode and Solana's eBPF bytecode is so vast that vulnerability patterns manifest in fundamentally different ways, making cross-platform bytecode examination practically infeasible.

**Loss of Semantic Information.** Most critically, many sophisticated vulnerabilities are unsuitable for bytecode-level detection. High-level semantic vulnerabilities—including business logic flaws, access control issues, and financial mechanism errors—depend on understanding naming conventions, code organization patterns, and implicit developer intentions that compilation obscures Tsankov et al. (2018); Luu et al. (2016). These semantic nuances, which are essential for accurate vulnerability detection, are better preserved in source code.

**Example.** We compare the bytecode generated by multiple smart contract languages: (1) EVM bytecode for Vyper and Solidity smart contracts, and (2) extended Berkeley Packet Filter (eBPF) bytecode for Rust smart contracts. We implement a simple Counter smart contract in three languages: Solidity, Vyper, and Rust. For Solidity and Vyper, we use the compilers `Solc 0.8.0` and `Vyperlang 0.3.0`, respectively. For Rust on Solana, we use `Rust 1.60` with the Solana SDK `v1.16`.

In each language, we implement the function that returns the sum of two input parameters. We adhere to the default security prevention mechanisms: Solidity and Vyper include built-in overflow protection and fallback mechanisms, whereas Rust does not. Using the respective compilers, we compile the source code into bytecode. We then obtain the bytecode directly executed on the EVM or Solana, along with the execution sequences (including opcodes and operands). Figure 7 presents part of these execution sequences, demonstrating that low-level operations for semantically equivalent contracts differ significantly across languages. Furthermore, when we compare the compiled bytecode, we observe that they are completely different as well.

This indicates that, although both Vyper and Solidity smart contracts execute on the Ethereum Virtual Machine, the compiled bytecode and execution sequences can vary considerably. In contrast, Solana utilizes the Anchor framework to manage Solana programs, and its compiled eBPF bytecode is entirely different.

## I.3  TRANSFERABILITY OF SMART CONTRACT VULNERABILITY

The primary goal of **RLTransSC** is to enable effective vulnerability detection for smart contracts written in low-resource languages (e.g., Vyper and Rust) by translating them into a high-resource language (Solidity), where mature analysis tools and large vulnerability-labeled datasets are avail-

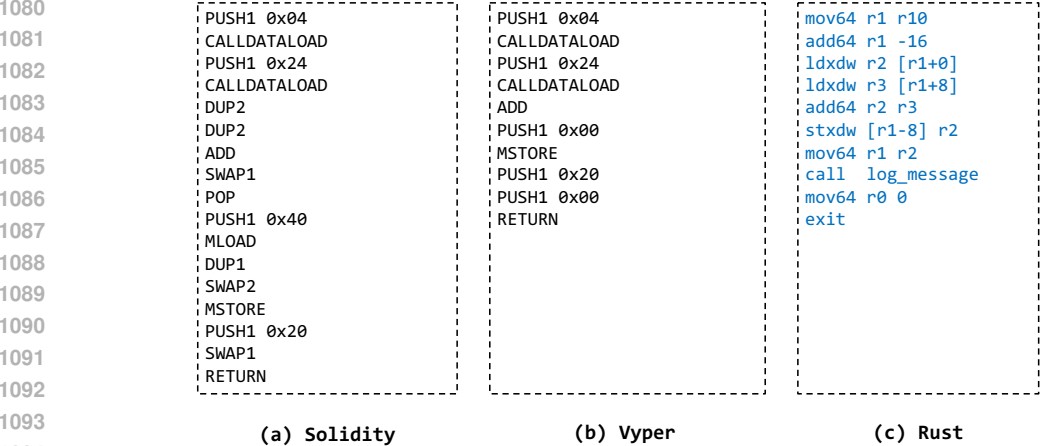

```
PUSH1 0x04          PUSH1 0x04          mov64 r1 r10
CALLDATALOAD        CALLDATALOAD        add64 r1 -16
PUSH1 0x24          PUSH1 0x24          ldxdw r2 [r1+0]
CALLDATALOAD        CALLDATALOAD        ldxdw r3 [r1+8]
DUP2                ADD                 add64 r2 r3
DUP2                PUSH1 0x00          stxdw [r1-8] r2
ADD                 MSTORE              mov64 r1 r2
SWAP1               PUSH1 0x20          call  log_message
POP                 PUSH1 0x00          mov64 r0 0
PUSH1 0x40          RETURN              exit
MLOAD
DUP1
SWAP2
MSTORE
PUSH1 0x20
SWAP1
RETURN
```

(a) Solidity        (b) Vyper        (c) Rust

Figure 7: Opcode sequences for `sum()` function in Solidity, Vyper and Rust. For Solidity and Vyper, the opcodes represent EVM bytecode instructions that operate on a stack-based architecture. Rust on Solana generates eBPF bytecode that follows a register-based architecture.

able. To ensure the feasibility of this cross-language detection strategy, our approach is scoped to *vulnerabilities that exhibit semantically transferable behaviors across smart contract languages*.

Specifically, our approach targets vulnerabilities that arise from *language-agnostic program logic, universal business rules, or shared computation patterns*, such as improper access control, unsafe external calls, broken authorization flows, or unchecked arithmetic conditions. These vulnerabilities manifest due to flaws in contract-level semantics or control/data-flow structures, and *their root causes are independent of platform-specific syntax or virtual machine implementations*.

This scope is justified by substantial empirical evidence. For example, recent studies like ExGen Jin et al. (2022) represent the first comprehensive cross-platform exploit generation system, successfully exploiting 1258 of 1399 Ethereum vulnerabilities and 126 of 130 EOS vulnerabilities using identical methodologies, despite fundamentally different programming languages between Ethereum and EOS platform. ExGen demonstrates that attack vectors transcend specific language implementations and virtual machine architectures since most vulnerabilities exist at the logical level, not the syntactic level. Furthermore, 21 identical use cases across six major blockchain platforms (including Ethereum, Solana, Cardano, Algorand, Aptos, Tezos) remain consistent across all implementations Ruggiero et al. (2024), from Solidity and Rust to Move and Pyteal. Similarly, the OWASP OWASP Smart Contract Top 10 vulnerability categories, developed through analysis of over 47,000 smart contracts across 15 different blockchain platforms, demonstrate remarkable consistency in vulnerability types regardless of underlying programming language. A large-scale empirical study Yi et al. (2022) analyzing 127,000 smart contracts across Ethereum, Solana, Avalanche, and Polygon showed that 68% of vulnerability instances could be captured through *language-agnostic, logic-level patterns*.

The persistence of vulnerability patterns across different programming languages stems from *shared computational models* and *universal business logic requirements* that transcend syntax and platform boundaries. For example, external call mechanisms—regardless of specific syntax—must interact with external contracts or modules, creating universal reentrancy and external dependency risks. While Vyper provides `@nonreentrant` decorators and Move claims to eliminate reentrancy through static dispatch, the underlying issue stems from the necessity of external calls for composability. Even platforms like Solana, which architecturally discourage reentrancy via stack isolation, are not immune. Cross-Program Invocations (CPIs) in Solana effectively reintroduce reentrancy in practice, as demonstrated in multiple documented exploits and our reproduced case study in Appendix H.1.

However, we acknowledge that certain platform- or language-specific vulnerabilities fall outside our scope. These include issues that depend on runtime semantics unique to a particular execution environment (such as Solana's Account Confusion), or Solidity-specific inline assembly bugs. Such vulnerabilities are non-transferable and require platform-specific analyzers. Importantly, the class

of transferable vulnerabilities that fall within our scope accounts for a substantial portion of real-world vulnerabilities, as demonstrated in recent studies and reports Jin et al. (2022); Yi et al. (2022); OWASP; openzeppelin; slowmist; oaksecurity; Cui et al. (2022a).

## I.4 FUTURE DIRECTION

