# OpenReview forum: "Bridging Language Barriers in Smart Contract Security: A Reinforcement Learning-Based Translation Approach"
_ICLR.cc/2026/Conference — ICLR 2026 Conference Withdrawn Submission_

### Official Review · Reviewer_8GU3 · 2025-10-26

**Soundness:** 3
**Presentation:** 3
**Contribution:** 2
**Rating:** 4
**Confidence:** 3

**Summary:**

This paper introduces a novel RL-based framework to translate smart contracts from low-resource languages such as Vyper and Rust into Solidity. The solution follows a two-stage approach: i) Finetuning an LLM with PPO that translates low-resource contracts to high-resource contracts. ii) Training a vulnerability detection model on Solidity dataset. Translated contracts are classified using this model and experimental promising results are promising.

**Strengths:**

- The problem is critical and the proposed solution is well-motivated.
- Translation model is optimized by taking both AST graph similarity and also functional correctness.
- The paper is well-structured and clear.
- Experiments are extensive and results are promising across models and various vulnerability types.

**Weaknesses:**

- The reward function in translation rewards functional and structural similarity. What if the translated contract has vulnerabilities? Can the translation unit fix those vulnerabilities implicitly? Because the test case execution reward seems to enforce so. This should be clarified.
- Reward scores for test case execution are manually set.
- For AST similarity reward and test-time GVR AST similarity, authors use different similarity metrics (WL kernel and GraphCodeBERT). The mismatch should be justified and explained.
- More details on the construction of the parallel dataset are needed.

**Questions:**

- Does the parallel dataset contain vulnerabilities? What is the impact of its size on results? These should be clarified.
- What is the impact of GVR on performance?

---

### Official Review · Reviewer_mMf7 · 2025-10-30

**Soundness:** 3
**Presentation:** 3
**Contribution:** 2
**Rating:** 2
**Confidence:** 4

**Summary:**

This paper presents RLTransSC, a reinforcement learning-guided framework for translating smart contracts from low-resource languages (Vyper, Rust) to high-resource languages (Solidity) while preserving vulnerabilities. The key innovation is using PPO to fine-tune LLMs with a custom reward function that encourages both functional correctness and structural similarity, enabling effective vulnerability detection in under-resourced blockchain ecosystems.

**Strengths:**

Originality: This is the first work to apply reinforcement learning for fine-tuning LLMs to translate smart contracts from low-resource languages to high-resource languages, enabling the reuse of existing vulnerability detection tools. The problem formulation of vulnerability-preserving translation is novel and the use of PPO with differential fuzzing represents a creative combination of techniques from code intelligence and security domains.

Quality: The technical approach is sound, featuring a well-designed multi-component reward function that balances structural preservation and functional correctness. The experimental evaluation is comprehensive, covering multiple models, hyperparameter studies, and comparisons with both direct detection baselines and supervised fine-tuning approaches across four vulnerability types.

Clarity: The paper is clearly written with well-motivated problem formulation, intuitive framework design, and thorough methodology exposition. The three-stage pipeline is easy to follow, and the case study effectively illustrates the end-to-end workflow.

Significance: The work addresses a practical challenge in blockchain security by enabling vulnerability detection for low-resource smart contract languages, though the impact is somewhat limited to the specific domain of blockchain security and the four vulnerability types studied.

**Weaknesses:**

1. While the paper claims to preserve vulnerabilities during translation through prompt instructions, this relies heavily on LLM behavior without theoretical analysis or systematic empirical validation. Only one case study is provided in Appendix H.1, which is insufficient to support this core contribution. The paper lacks quantitative statistics on preservation rates across all test samples, and provides no analysis of failure cases where vulnerabilities might be accidentally fixed. Without direct measurement of preservation rates, the claim remains inadequately substantiated. Additionally, the relative importance of AST similarity versus test execution in preserving vulnerabilities is not empirically validated.

2. The paper does not specify the proportion of vulnerable versus safe contracts in the training and test datasets, making it impossible to assess whether the evaluation is balanced. More critically, the F1-score metric cannot distinguish between detection failures and cases where vulnerabilities were inadvertently fixed during translation. The paper is advised to report explicit preservation rates showing how many vulnerabilities in source contracts remain detectable in translated contracts, and analyze cases where vulnerabilities are lost or new ones are introduced during translation.

3. The paper lacks ablation studies isolating the contribution of each reward component. A systematic hyperparameter search or sensitivity analysis for reward weights would strengthen the work. Additionally, exploring learned reward weighting schemes (like RLHF) could potentially improve upon the manual design.

4. Although the authors state plans to release code, models, and datasets publicly, none are provided in supplementary materials or anonymous repositories (such as anonymous.4open.science). The lack of available artifacts makes it difficult to verify results, which is particularly important given the novel claims about vulnerability preservation.

**Questions:**

Could the translation process introduce new vulnerabilities due to language-specific semantic differences? For example, when translating from Vyper (which has overflow protection) or Rust (with strict ownership rules) to Solidity, could safety guarantees be lost? A concrete case: Vyper's assert causes state reversion without refunding gas, while Solidity's require refunds remaining gas—if translated incorrectly, this could enable gas-based attacks or DoS vulnerabilities. Please provide concrete examples of how your approach handles such semantic mismatches and ensures security properties are preserved.

---

### Official Review · Reviewer_fboT · 2025-10-31

**Soundness:** 2
**Presentation:** 3
**Contribution:** 2
**Rating:** 4
**Confidence:** 3

**Summary:**

The paper proposed RLTransSC, a framework that utilizes reinforcement learning-guided LLM fine-tuning to translate smart contracts from low-resource languages into a high-resource language (i.e., Solidity), enabling the application of mature security tools and rich labeled datasets. Evaluated on 1k+ smart contracts, RLTransSC showed improvement on performance of semantic-preserving translation and transferability of vulnerability detection.

**Strengths:**

**Originality**
The paper proposed novel approach of translating smart contracts from low-resource language into high-resource language to reuse vulnerable detection techniques

**Quality**
The paper made extensive experiments on various LLMs and different finetuning methods, showing quantitative results of program translation and vulnerability detection performance with multiple metrics.

**Clarity**
The paper is well-written with description of workflow and RL reward definition.

**Significance**
The paper proposed a framework which can be used to help bug-finding for smart contracts written with low-resource languages, though no new vulnerability directly reported.

**Weaknesses:**

**ground-truth dataset construction**
Since the RL-based approach utilizes similarity between translated and ground-truth Solidity contracts as part of rewards. It's crucial to ensure the quality of ground-truth dataset. However, the back translation approach seems to be simply calling more-advanced LLMs without further details about cost and successful rate. The differential testing phase may find shallow discrepancy easily but cannot provide formal guarantee or verification of the equivalence even if line coverage is high. I am not sure if it's possible to use traditional program translation/synthesis technique to construct more reliable ground-truth dataset, or is it fundamentally possible to translate Vyper/Rust smart contracts into Solidity while keeping full semantic. The authors may consider elaborate more on this.

**motivation of fine-tuning small LLM to improve translation performance**
Since the ground-truth dataset consisting of 1k+ smart contracts pairs can be (automatically) constructed by querying more advanced LLM (including GPT-4.1, Llama-3.2, Deepseek-R1, Claude-Sonnet-4, Gemini-2.5-Flash) and applying differential testing. The process itself seems to be simple and good enough for the translation task. It's hard for me to see the motivation of fine-tuning smaller LLM to achieve minimal improvement. The authors may consider mentioning the financial cost and successful rate of building the ground-truth dataset versus fine-tuning LLMs.

**cross-function program translation and vulnerability detection**
The paper does not mention whether the framework translate the smart contract at function-level or as a whole, which is important to estimate the difficulty of program translation task. In addition, it seems AmeVulDetector models the vulnerability detection as function-level binary-classification problem, which is not a proper setting for many vulnerability types. As discussed in the paper below, the high F1 score of machine learning for vulnerability detection (ML4VD) may not reflect real performance of vulnerability detection. Although the paper's main contribution is not directly related to ML4VD, it would be good to see some discussion on cross-function program translation and vulnerability detection.

Top Score on the Wrong Exam: On Benchmarking in Machine Learning for Vulnerability Detection (ISSTA '25)

**Questions:**

1. What's the ratio of smart contracts generated by back translation passing the differential testing check?
2. What's the financial cost of constructing ground-truth dataset with advanced LLMs and fine-tuning smaller LLMs, respectively?

---

### Official Review · Reviewer_PkPq · 2025-11-01

**Soundness:** 3
**Presentation:** 3
**Contribution:** 3
**Rating:** 6
**Confidence:** 4

**Summary:**

This paper introduces the idea of leveraging high resource smart contract language Solidity with good vulnerability detection tools to improve the vulnerability detection in low-resource smart contract languages such as Vyper and Rust. The main idea is to utilize a reinforcement learning (RL) approach to train a translation model to map the smart contracts written in Vyper or Rust into Solidity smart contracts.

**Strengths:**

I found the idea of leveraging Solidity smart contracts and their associated vulnerability detection tools to guardrail those smart contracts written in other languages such as Vyper and Rust practical though this idea has been around and different research efforts focus on different approaches to leveraging Solidity based vulnerability detection tools.

I also like the proposal of using RL to train a translation model to translate the low-resource smart contract programs in Vyper or Rust to Solidity. One of the main design ideas of the proposed RL model is the composite reward function which combines AST based rewards for structured similarity and test-case execution reward for semantic correctness with KL divergence regulation term to control the drastic derivation of fine-tuned model from the original model.

The experimental evaluation is performed on using four LLM models to perform RL-training for transaction (Vyper-to-Solidity and Rust-to-Solidity). Table 1 shows the high performance gain in transaction evaluation and Table 2 shows the high vulnerability detection performance of all four models with RL training for transaction.

**Weaknesses:**

The paper could benefit from better elaboration on the following aspects:

(1)  During parallel dataset construction, those failing verification step will be excluded from the dataset. Hence, only 1102 pairs from Vyper-Solidity and 1067 pairs for Rust-Solidity. Evaluation is done by randomly sampling 50 pairs from the verified datasets. Could you elaborate whether this specific dataset treatment and the test sampling approach are the key reasons for the high performance results in both Table 1 and Table 2.

(2) The paper did not provide any ablation study showing the role of parallel dataset construction, the evaluation test data construction, the RL training of translation with both composite reward function vs one of the reward plus KL-divergebce and how each impact on the vulnerability detection accuracy.

(3) Discuss sensitivity boundary scenarios and how the proposed method behave instead of "assuming the translation is accurate".
For example, when the Vyper or Rust smart contract being tested are similar to those that were filtered out at the parallel dataset construction stage, hence the RL translation model has no knowledge of how to translate them into correct solidity smart contract formats.

(4) The setting of hyper parameters seemed to be tuned manually (Section 4.3). Some discussion on scenarios when such fine-tuned settings may no longer work.

**Questions:**

See questions in the Weakness section.

---

### Note · Authors · 2025-11-21

**Comment:**

We would like to thank all the anonymous reviewers for their time in reviewing this paper. We will carefully use the suggestions to improve this work in the future.

**Withdrawal Confirmation:**

I have read and agree with the venue's withdrawal policy on behalf of myself and my co-authors.